# The Impact of Trust on Chinese Consumers’ Acceptance of Meat Substitutes: The Mediating Role of Perceived Benefits and Perceived Risks, and the Moderating Role of Consumer Knowledge

**DOI:** 10.3390/foods14040669

**Published:** 2025-02-16

**Authors:** Muhabaiti Pareti, Junsong Guo, Nadire Abudurofu, Qiankun Liu, Abulizi Bulibuli, Maurizio Canavari

**Affiliations:** 1College of Economics and Management, Xinjiang Agricultural University, Nongda East Road 311, Urumqi 830052, China; 18997932247@163.com (J.G.); 18199861480@163.com (N.A.); 2Department of Agricultural and Food Sciences, Alma Mater Studiorum-University of Bologna, Viale Guiseppe Fanin 50, 40127 Bologna, Italy; qiankun.liu2@unibo.it (Q.L.); maurizio.canavari@unibo.it (M.C.); 3School of Business and Administration, Xinjiang University of Finance & Economics, Beijing Middle Road 449, Urumqi 830012, China; abliz@xjufe.edu.cn

**Keywords:** consumer trust, consumer benefit, consumer risk, consumer knowledge, consumer acceptance, meat substitutes

## Abstract

In recent years, meat substitutes have become a prominent global topic in academic research. As one of the world’s most populous countries, China may increasingly consider meat substitutes as a new dietary option to meet the growing consumer demand. The potential acceptance of meat substitutes by consumers is a critical foundation for the industry’s development, as it directly influences it’s success. Moreover, consumer trust in meat substitutes plays a pivotal role in shaping this potential acceptance. Consequently, this study aims to analyze the impact of consumer trust on potential acceptance. Additionally, it incorporates the mediating roles of perceived benefits and perceived risks, as well as the moderating effect of consumer knowledge. A questionnaire survey was administered to 2647 Chinese consumers and a moderated dual-mediation model, based on the Bootstrap method, was employed to investigate the effects of consumer trust on the potential acceptance of meat substitutes. The study subsequently validated the mediating roles of perceived benefits and perceived risks, along with the moderating role of consumer knowledge. The findings indicate that consumer trust positively influences the potential acceptance of meat substitutes. Furthermore, consumer trust enhances perceived benefits while simultaneously reducing perceived risks. It affects consumer acceptance both directly and indirectly, through the mediating variables of perceived benefits and perceived risks. Consumer knowledge moderates the relationships between consumer trust, perceived benefits, and perceived risks, but does not significantly moderate the relationship between consumer trust and consumer acceptance.

## 1. Introduction

With the rapid population growth, consumer demand for meat products is steadily increasing [1]. This rising demand has exerted significant pressure on the environment and grassland ecosystems [2,3]. The conventional meat industry is a major contributor to greenhouse gas emissions. To mitigate greenhouse gas emissions from animal husbandry, enhance animal welfare, and conserve energy and water, meat substitutes such as plant-based protein meat and cell-based meat are emerging as potential strategies for future food development across various countries. Cultured meat has already been introduced to dining tables in Singapore [4]. A survey conducted by Anne Cardin Jacobs and other scholars on German consumers’ attitudes towards artificial meat shows that the majority of respondents believed that the traditional meat industry was currently facing ethical and environmental problems and that reducing meat consumption was likely to be a solution to some of these problems [5]. The proportion of existing plant-based meat alternatives that meet the needs of Belgian consumers has significantly increased, from 44% in 2019 to 51% in 2020. People’s attention to issues related to animal husbandry, especially environmental issues, has significantly increased [6]. Chinese consumers’ attitudes towards meat consumption are primarily focused on private interests such as nutrition, safety, and quality, followed by public interests.

Given the constraints of carbon peak and carbon neutrality targets, meat substitutes could serve as supplementary sources of meat supply in China. Consequently, cultured meat and plant-based meat may become viable alternatives for Chinese consumers in the future. In recent years, plant-based protein meat and cultured meat have garnered public attention, becoming a global topic of interest among scientists, media, and consumers [7]. To meet the growing demand for meat products, major food manufacturing enterprises and research institutions in China are actively investing in research and development of plant-based and cell-based meat substitutes [8]. Scholars have investigated the acceptance and purchase intentions of Chinese consumers toward meat substitutes, exploring the main influencing factors [9,10,11]. The results showed that Chinese consumers prefer vegetarian and plant-based meat [12]. Scholars have analyzed how consumer characteristics—such as knowledge, perception, attitudes, and framing—impact consumer acceptance or purchase intentions [13]. Consumers’ trust in the manufacturing technologies and regulatory processes of meat substitutes may significantly influence their acceptance [14]. Beyond technological advancements, consumer acceptance remains the most critical factor for the large-scale commercialization of meat substitutes in the Chinese market.

Presently, Chinese consumers are informed about meat substitutes through media platforms such as TikTok, Kwai, and various television programs. Despite media coverage highlighting the environmental benefits [2], resource conservation, nutritional value [15], and positive impact on animal welfare of plant-based meat [16], many consumers still exhibit a cautious or skeptical attitude. This skepticism is due to the relatively short history of meat substitutes, along with underdeveloped manufacturing and regulatory technologies, and concerns related to taste [17], authenticity [18], price [10], and other factors [7]. Thus, in addition to developing mature manufacturing technologies, achieving high consumer acceptance is essential for the successful integration of meat substitutes into the market.

While plant-based meat has already been commercialized in China, cultured meat remains in the experimental research stage; limited by manufacturing, regulatory, and legislative challenges. Regardless of whether the meat is plant-based or cultured, consumer acceptance is the critical driver for the growth of these novel industries. Understanding consumers’ acceptance of these meat substitutes and the factors influencing their attitudes is vital for diversifying and expanding the future meat market.

Consumer acceptance toward meat substitutes is influenced by their trust in manufacturing companies, regulatory agencies, and media sources. Additionally, perceived benefits and perceived risks, alongside consumer trust, significantly affect purchasing intentions. Consumer knowledge about meat substitutes enhances their perception of benefits and risks, encouraging informed decision-making rather than blind reliance on media. This study explores the intricate relationships among consumer trust, perceived benefits, perceived risks, consumer knowledge, and purchase intentions; synthesizing existing research to construct a theoretical model.

The primary aim of this study is to assess consumer acceptance of meat substitutes and understand the impact of consumer trust on acceptance. Additionally, the study seeks to explore the mediating roles of perceived benefits and perceived risks, as well as the moderating effect of prior consumer knowledge. The structure of this paper is as follows. Section 2 sorts out the literature and research hypotheses, focusing on the influence of consumer trust in meat substitute technologies and regulatory systems on purchase intentions; alongside the mediating and moderating roles of perceived benefits, perceived risks, and consumer knowledge. Section 3 details data collection and research methodology. Section 4 presents data analysis and results. Section 5 discusses the findings, and Section 6 concludes the research, highlighting the limitations and future research directions.

## 2. Literature Review and Hypothesis Development

Beyond trust, consumers as rational economic agents evaluate the potential benefits and risks associated with products before making purchase decisions. Consequently, perceived benefits and perceived risks act as mediating factors between consumer trust and purchase intentions [19,20]. Wang et al. [21] have also discussed the role of consumer knowledge in shaping consumer trust. Amarullah et al. [22] demonstrated that consumer knowledge moderates the relationship between trust and purchase intention. Furthermore, some studies have examined how knowledge moderates the relationship between consumer perception and purchase intention [23].

### 2.1. Trust and Consumer Acceptance

Trust is a psychological state of consumers, in which there are positive expectations for a new product [24]. Here, trust is a multidimensional concept which refers to consumers’ trust in manufacturing companies, regulatory agencies, public media, etc. [25]. Scholars have identified consumer trust as a fundamental factor influencing the acceptance of novel foods [14,26,27]. A lack of trust leads to a corresponding decline in purchase intentions [28]. Plant-based and cultured meat represent the latest advancements in meat substitutes, with media serving as the primary source of information for most consumers. Consequently, trust in media and regulatory agencies is critical to consumer acceptance of these products. Trust in the food industry enhances acceptance of new food products [29]. The literature indicates that trust has a direct impact on consumer purchasing intentions [30]. In previous studies, consumer acceptance was defined as consumer purchasing and usage intention [26,31]. Based on this understanding, we propose the following hypothesis:

**Hypothesis 1 (H1).** 
*Trust has a positive impact on consumers’ acceptance.*


### 2.2. The Mediating Role of Perceived Risk and Benefit Between Trust and Consumer Acceptance

The perceived benefits emphasized by consumers include the advantages of meat substitutes in environmental and ecological protection, animal welfare, and nutritional balance. At the same time, the perceived risks of taste, health, and nutrition brought by meat substitutes are also the main aspects that affect consumers’ purchasing tendencies. Whether consumers purchase meat substitutes depends to some extent on the interaction between their perceived benefits and perceived risks, based on their trust in relevant manufacturers, regulatory agencies, and media. Scholars have emphasized the mediating role of perceived risk and perceived benefit between consumer trust and acceptance. They noted that, while trust directly influences consumer acceptance, it also exerts an indirect effect through perceived risk, which serves as a mediating factor [32,33]. Some scholars consider perceived risk as a negative mediator between consumer trust and consumer purchase intention [34,35]. However, the mediating role of perceived benefits has often been overlooked. Bronfman et al. [36] addressed this gap by incorporating perceived benefits into their theoretical model, which encompasses trust, perceived risk, perceived benefit, and consumer acceptance. Drawing from this theory, we propose two hypotheses:

**Hypothesis 2a (H2a).** 
*Perceived risk has a negative mediating role between trust and consumer acceptance.*


**Hypothesis 2b (H2b).** 
*Perceived benefit has a positive mediating role between trust and consumer acceptance.*


### 2.3. The Moderating Role of Consumer Knowledge

Consumer knowledge about the technology and regulatory framework surrounding meat substitutes significantly impacts purchasing behavior [37]. In the context of novel food technologies, the evaluation of potential benefits and risks is heavily influenced by the consumer’s knowledge level [29,38]. When knowledge is limited, the connection between trust and perceived risks or benefits becomes more pronounced [32]. Building on this understanding, we assert that consumer trust in manufacturing companies and regulatory agencies is moderated by the level of consumer knowledge about meat substitutes. Specifically, consumer knowledge influences the perception of risks and benefits associated with these products. Bronfman et al. [36] emphasized that knowledge is pivotal in modulating the direct impact of trust on consumer acceptance. When the public has limited knowledge, trust in institutions and media strongly influences acceptance. As knowledge increases, trust becomes less critical, and consumers make more rational evaluations of risks and benefits. Therefore, we hypothesize:

**Hypothesis 3a (H3a).** 
*Consumer knowledge moderates the relationship between consumer trust and consumer acceptance.*


**Hypothesis 3b (H3b).** 
*Consumer knowledge moderates the relationship between consumer trust and consumer perceived risk.*


**Hypothesis 3c (H3c).** 
*Consumer knowledge moderates the relationship between consumer trust and consumer perceived benefit.*


The theoretical relationships among consumer trust, perceived risk, perceived benefit, consumer knowledge, and consumer acceptance are illustrated in Figure 1.

## 3. Materials and Methods

### 3.1. Data Collection

This study employs a questionnaire survey to test the research hypotheses. The questionnaire comprises four main sections: the first part captures basic demographic information, including gender, age, education level, and income. The second section assesses consumers’ awareness of the manufacturing process, industry trends, and pricing of meat substitutes. The third section examines consumers’ trust in the manufacturing processes of meat substitutes, the regulatory technologies of relevant authorities, media coverage reliability, and the transparency of traceability systems. The fourth section explores perceived benefits, such as environmental and ecological protection, energy and water conservation, animal welfare, nutritional balance, and consumer well-being. Additionally, it addresses perceived risks, including unnatural taste, uncertainties about long-term health effects, negative impacts on traditional meat industries and workers, and monopolistic practices of large producers. The final section measures consumer acceptance; specifically, their willingness to try, consume, purchase, and recommend meat substitutes. Given time and budget constraints, an online survey method was chosen to minimize data input errors associated with face-to-face interviews. The survey was conducted using the professional questionnaire platform Wenjuanxing (https://www.wjx.cn/ (accessed on 30 June 2024)). The authors registered on the Wenjuanxing platform and uploaded the questionnaire to generate a digital survey link. The link was disseminated through major WeChat groups and online forums, inviting participants to complete the survey. The survey will be conducted from April to August 2024. A total of 2647 questionnaires were distributed, and 2241 valid responses were retained after thorough review and removal of invalid entries.

### 3.2. Measures

#### 3.2.1. Trust

A five-point Likert scale was employed to evaluate consumers’ perceptions of industry technology maturity, regulatory system robustness, media coverage reliability, and the transparency of traceability systems for meat substitutes. The Cronbach’s alpha coefficient for this scale is 0.793, indicating acceptable reliability.

#### 3.2.2. Knowledge

The scale consisted of four items assessing various dimensions of consumer knowledge, including awareness of meat substitute types, industry development, manufacturing technology, and pricing. Each item reflects consumers’ knowledge levels across different aspects of meat substitutes. The Cronbach’s alpha coefficient for this scale is 0.780, indicating adequate reliability.

#### 3.2.3. Perceived Benefit

Five items were developed to measure perceived consumer benefits, encompassing environmental protection, energy and water conservation, reduced animal suffering, balanced nutrition, and lower infectious disease risk. The Cronbach’s alpha coefficient for this scale is 0.902, indicating high reliability.

#### 3.2.4. Perceived Risk

The scale included four items assessing perceived risks: uncertainty about long-term health effects, unnatural taste, negative impacts on traditional animal husbandry, and potential consumer welfare losses from monopolistic practices. The Cronbach’s alpha coefficient for this scale is 0.785, indicating sufficient reliability.

#### 3.2.5. Consumer Acceptance

Consumer acceptance was measured using a single item: “I am willing to purchase meat substitutes”. 

Table 1 contains the reliability test results of the above five variables.

## 4. Results

This study first performs descriptive statistical analysis on consumer demographics and model-related variables, followed by hypothesis testing on the mediating roles of perceived risk and perceived benefit. Subsequently, it examines the moderating model of consumer knowledge. The research primarily employs the Bootstrap method for mediating and moderating effect testing, as recommended by Hayes, using SPSS 27 as the analytical tool.

### 4.1. Descriptive Analysis

The main control variables include: gender (1288 males and 953 females, representing 57.47% and 42.53%, respectively); age (18–30 years old: 40.96%, 31–40 years old: 20.04%, 41–50 years old: 16.82%, 51–60 years old: 11.33%, over 60 years old: 10.84%); educational background (junior high school or below: 13.16%, high school or technical school: 25.08%, junior college or university degree: 35.03%, bachelor’s degree or above: 26.73%); and annual household income (below ¥20,000: 8.57%, ¥20,000–¥50,000: 18.3%, ¥50,000–¥100,000: 37.62%, ¥100,000–¥200,000: 24.5%, above ¥200,000: 11.02%). Additionally, consumer proportions by residence type are 17.09% from large cities, 30.57% from small and medium-sized cities, 23.34% from small towns, and 29% from rural areas. Regional distribution shows that 54.22% of consumers are from the eastern region, with the remaining 45.78% from central and western regions.

According to statistical analysis results (Table 2), overall consumer purchase intentions, trust levels, knowledge levels, perceived risks, and perceived benefits are above the median. The mean purchase intention among consumers is 2.85, while perceived risk and perceived benefit scores are relatively high, with means of 3.24 and 3.18, respectively. However, consumer trust and knowledge levels related to meat substitutes are comparatively low, with means of 2.82 and 2.81, respectively.

Table 3 demonstrates that consumer acceptance correlates most strongly with consumer trust (r = 0.635), followed by perceived benefits (r = 0.629), consumer knowledge (r = 0.624), and perceived risks (r = −0.620).

### 4.2. The Mediating Effect of Perceived Risk and Perceived Benefit

We used Hayes’ PROCESS macro for data analysis, defining consumer purchase intention as the dependent variable, consumer trust as the independent variable, and perceived risk and perceived benefit as mediating variables. The self-sampling frequency was set to 5000, and the detailed results are shown in Table 4. Findings indicate that consumer trust has a significant positive effect on consumer acceptance (β = 0.3197, *p* < 0.001), confirming Hypothesis 1. Trust also significantly reduces perceived risk (β = −0.8005, *p* < 0.001) and significantly enhances perceived benefit (β = 1.4736, *p* < 0.001). Perceived risk (β = −0.2268, *p* < 0.001) negatively affects consumer acceptance, while perceived benefit (β = 0.2577, *p* < 0.001) has a significant positive impact on consumer acceptance. These results suggest that perceived risk and perceived benefit partially mediate the relationship between trust and consumer acceptance.

In the bootstrap indirect effect analysis of perceived risk and perceived benefit (Table 5), the indirect effect of perceived risk on the relationship between trust and consumer acceptance is 0.2041; and for perceived benefit, it is 0.2588. The 95% bootstrap confidence intervals for both variables do not include zero, confirming that perceived risk and perceived benefit mediate the influence of trust on consumer acceptance.

### 4.3. Moderating Effect of Consumer Knowledge

There is a notable correlation between consumer knowledge and consumer trust, perceived risk, perceived benefit, and consumer acceptance. To further investigate these relationships, a model was developed with consumer acceptance as the dependent variable, consumer trust as the independent variable, perceived risk and perceived benefit as mediators, and consumer knowledge as a moderator. The self-sampling frequency was set to 5000, and the detailed results are provided in Table 6. According to the regression analysis, consumer trust has a significant negative effect on perceived risk (β = −0.44, *p* < 0.001) and a significant positive effect on perceived benefit (β = 0.46, *p* < 0.001) and consumer acceptance (β = 0.32, *p* < 0.001). Furthermore, knowledge negatively impacts perceived risk (β = −0.40, *p* < 0.001) and has a significant positive impact on perceived benefit (β = 0.46, *p* < 0.01) and consumer acceptance (β = 0.22, *p* < 0.001).

Additional analysis reveals that the interaction term between trust and knowledge significantly affects perceived risk (β = −0.12, *p* < 0.001) and perceived benefit (β = −0.10, *p* < 0.001) but does not significantly impact consumer acceptance (β = −0.01, *p* > 0.05).

Based on the means of the two continuous variables, trust and knowledge, the parts that are more than one standard deviation above the mean value are classified as high trust and high knowledge, and the parts that are more than one standard deviation below the mean are classified as low trust and low knowledge. This study chose to convert the continuous variables of trust and knowledge into binary variables, mainly for the purpose of simplifying the analysis and making the relationships between variables more interpretable in certain analyses. The implications of this binary conversion are two-fold. On the one hand, it provides a clear and easy-to-understand way to present the data, especially for readers who may not be familiar with more complex statistical analyses. For example, in Figure 2, it becomes immediately apparent how different levels of trust or knowledge are related to other variables. On the other hand, this study found that some information may be lost in the process of converting continuous variables to binary ones. However, we believe that the benefits of simplification and enhanced interpretability in this stage of the analysis outweigh the potential loss of detail.

By categorizing consumer trust into high and low levels based on one standard deviation above and below the mean, the moderating effects of knowledge for both high- and low-knowledge consumers were analyzed separately. Figure 2 displays the specific moderating effects.

The regression coefficients for consumer knowledge on perceived risk (β = −0.44, *p* < 0.001) and perceived benefit (β = 0.46, *p* < 0.001) indicate that knowledge moderates the effect of trust on consumer acceptance through perceived risk and perceived benefit. Using the Bootstrap method to further validate this effect, a 95% confidence interval was calculated to determine the significance of knowledge levels on the trust–consumer acceptance relationship (see Table 7). Analysis shows that for respondents with high knowledge levels, the 95% confidence interval for perceived risk and perceived benefit does not include zero, confirming that trust influences consumer acceptance through perceived risk and perceived benefit. Similarly, for low-knowledge consumers, the confidence interval does not include zero, indicating that trust influences acceptance through perceived risk and benefit for both knowledge groups.

Table 7 shows that perceived benefits among high-knowledge consumers exhibit less variation with increasing trust than among low-knowledge consumers. This outcome illustrates the negative moderating effect of knowledge on the trust–perceived benefit relationship. Regarding the moderating effect of knowledge on trust and perceived risk, high-knowledge consumers experience a greater reduction in perceived risk with increased trust than low-knowledge consumers. In the low-knowledge group, the moderating effect of knowledge on the trust–perceived risk relationship is smaller than in the high-knowledge group.

## 5. Discussion

### 5.1. The Impact of Consumer Trust on Consumer Acceptance

The acceptance of meat substitutes is a topic of significant interest among researchers. The degree of trust consumers place in companies, institutions, and media associated with artificial meat affects not only their purchasing intentions directly but also indirectly through perceived risks and perceived benefits. Some scholars pointed out that the indirect influence of consumer trust on purchasing intent via these two mediating factors surpasses its direct effect [39]. In this study, we also found that the indirect effect of consumer trust on consumer acceptance through the mediating role of perceived risk and perceived benefit is slightly lower than its direct effect, and the regression coefficient size of the introduction effect is almost equivalent to that of the direct effect. This suggests that Chinese consumers tend to be rational, often assessing the potential advantages and risks of meat substitutes before deciding whether to buy them. Regarding the contribution of mediating effects, consumer perceived benefits have a slightly greater influence than perceived risks in mediating the relationship between consumer trust and purchasing intent, aligning with findings by scholars [40]. Whether it is a direct or indirect effect, consumer trust has a significant impact on consumers’ acceptance of meat substitutes. In terms of practical implications, it is necessary to improve the relevant regulatory system, strictly regulate market access, establish a sound market supervision mechanism, establish a traceability system for meat substitutes, carry out strict quality control, ensure the good quality and taste of meat substitutes, and enhance consumers’ trust in meat substitutes.

### 5.2. The Moderating Effect of Consumer Knowledge

When the moderating variable of consumer knowledge is introduced, the regulatory impact on consumer perceived risk shows a significant negative moderating effect, though the effect on the pathway between consumer trust and consumer acceptance remains insignificant. Consumers with higher knowledge levels experience greater perceived benefits [41] and lower perceived risks [42]. As consumer knowledge increases, they no longer uncritically accept media promotion, leading to a decline in the impact of media trust on perceived benefits [32]. With respect to perceived risk, higher consumer knowledge correlates with reduced risk perceptions based on media trust. Figure 2 illustrates that media influences the perceived benefits of low-knowledge consumers more strongly than those with high knowledge, while its impact on perceived risks is lower for low-knowledge consumers. This suggests that increased consumer trust has a more pronounced effect on perceived benefits for low-knowledge consumers, whereas it has a stronger effect on reducing perceived risks for high-knowledge consumers. Consumers with high levels of knowledge experience a significant decrease in perceived risk levels as their novice skills improve, while low-level consumers experience a higher increase in perceived benefits. In terms of practical implications, when relevant departments carry out science popularization activities, they emphasize more the cutting-edge knowledge of controllable risks of meat substitutes for consumers with high knowledge levels, reducing their worries. When conducting science popularization activities for consumers with low knowledge levels, they emphasize more the advantages of meat substitutes and the benefits they bring to consumers, improving their perceived benefits and ultimately increasing consumers’ acceptance of new food products such as meat substitutes.

### 5.3. Limitations and Future Research

This study uses Chinese consumers as a case study, to explore the influence of consumer trust on their acceptance of meat substitutes within a moderated dual mediation model. It examines the mediating roles of perceived risk and perceived benefit, as well as the moderating effect of consumer knowledge. This work further investigates the relationships among consumer trust, perceived benefits and risks, consumer knowledge, and acceptance; making a theoretical contribution to research on consumer behavior in the context of meat substitutes. However, there are certain limitations. First, due to the fact that most of the data used in this study are from online surveys, there may be some sample bias in the online survey data, which may overlook some groups that do not frequently use the internet. Therefore, the research results may have certain biases, which is a limitation of the generalizability of this study. Second, artificial meat is not yet commercialized in China, and plant-based meat substitutes have a low market penetration rate. Consequently, consumers’ knowledge, perceived benefits, and perceived risks of meat substitutes are largely informed by the media. Most consumers lack understanding of specific products such as cell-cultured meat. The meat substitutes currently available in the Chinese market are primarily plant-based products, such as egg substitutes and white meat, which may introduce biases in consumer responses to online surveys. Third, consumer acceptance in this study is defined solely by consumers’ willingness to purchase meat substitutes in the future, which differs from a broader concept of consumer acceptance. As meat substitutes gradually commercialize in the Chinese market, future research could enhance the understanding of consumer acceptance by incorporating behavioral indicators specific to meat substitutes.

## 6. Conclusions

This study utilizes survey data from Chinese consumers and employs a moderated dual mediation model to examine how consumer trust in meat substitutes influences acceptance. It explores the mediating roles of perceived benefits and perceived risks in the relationship between consumer trust and acceptance, and investigates how consumer knowledge moderates the effects of trust on these mediators.

Firstly, consumer trust in meat substitutes positively influences perceived benefits and acceptance, while negatively affecting perceived risks.

Secondly, perceived benefits and perceived risks mediate the effect of consumer trust on acceptance. The indirect effect of trust on acceptance, mediated by these variables, is stronger than the direct effect.

Lastly, consumer knowledge significantly moderates the relationships among trust, perceived benefits, and perceived risks. As consumer knowledge increases, the positive effect of trust on perceived benefits diminishes, whereas its negative effect on perceived risks becomes more pronounced.

## Figures and Tables

**Figure 1 foods-14-00669-f001:**
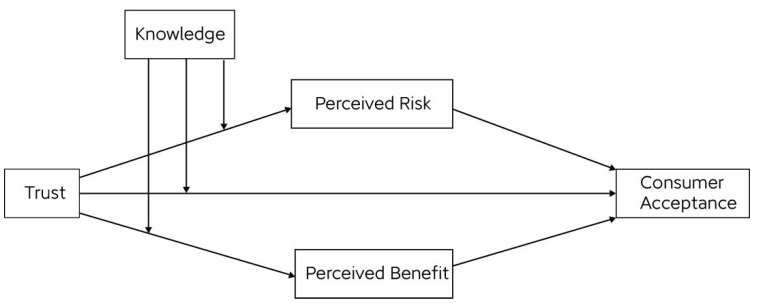
Hypothesized model of consumer acceptance of meat substitutes.

**Figure 2 foods-14-00669-f002:**
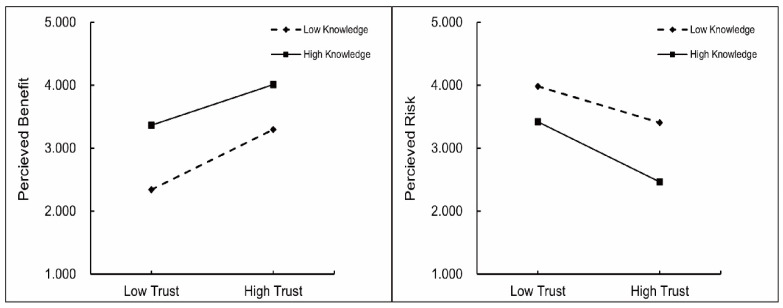
The moderating effect of knowledge on consumer perceived risk and perceived benefit.

**Table 1 foods-14-00669-t001:** Reliability and Validity Analysis.

	Items	Cronbach’s Alpha Value	CR	AVE
Willingness to buy	1 item	-	-	-
Consumer trust	4 items	0.793	0.866	0.618
Knowledge	4 items	0.780	0.860	0.604
Perceived benefits	5 items	0.902	0.923	0.631
Perceived risks	4 items	0.785	0.909	0.714

Since the CR values of all variables are greater than 0.7, it can be concluded that the convergence between each item and the latent variable is good.

**Table 2 foods-14-00669-t002:** Descriptive statistics of variables.

Variables	Min	Max	Mean	St. Deviation
Trust	1.00	5.00	2.8165	0.86702
Knowledge	1.00	5.00	2.8133	0.94257
Perceived risk	1.00	5.00	3.2402	0.93085
Perceived benefit	1.00	5.00	3.1898	0.89256
Consumer acceptance	1.00	5.00	2.8527	1.06767

**Table 3 foods-14-00669-t003:** Means, standard deviations, and correlations among variables.

Variable	CT	KW	PR	PB	CA
CT	1				
KW	0.786 **	1			
PR	−0.746 **	−0.746 **	1		
PB	0.818 **	0.825 **	−0.719 **	1	
CA	0.635 **	0.624 **	−0.620 **	0.629 **	1

Note: ** *p* < 0.01.

**Table 4 foods-14-00669-t004:** Regression analysis of major variables.

Regression Equation		Good-Fit Test	F-Test	Coefficient	*t*-Test
Dependent variable	Independent variable	R^2^	F	β	t
Perceived benefit	Trust	0.6691	4527.04	0.8421	67.28 ***
Perceived risk	Trust	0.5560	2803.53	−0.8005	−52.95 ***
Consumer acceptance	Trust	0.4601	635.54	0.3197	8.84 ***
Consumer acceptance	Perceived benefit			0.3073	9.12 ***
Consumer acceptance	Perceived risk			−0.2549	−9.14 ***

Note: *** *p* < 0.001.

**Table 5 foods-14-00669-t005:** The mediating effect of perceived risk and benefit.

Effect	Value	St. Err	95% C.I. Lower	95% C.I. Upper	Proportion
Total indirect effect	0.4629	0.0335	0.3958	0.5273	59.15%
Perceived benefit	0.2588	0.0303	0.1979	0.3173	33.07%
Perceived risk	0.2041	0.0231	0.1582	0.2483	26.08%

**Table 6 foods-14-00669-t006:** The moderating effect of consumer Knowledge.

Regression Equation		Fit Index		Coefficient	Significance
Dependent Variable	Independent Variable	R^2^	F	β	*t*
Perceived benefit	Trust	0.76	2420.26	0.46	27.07 ***
	Knowledge			0.46	29.30 ***
	Trust × Knowledge			−0.10	−9.05 ***
Perceived risk	Trust	0.64	1299.63	−0.44	−19.91 ***
	Knowledge			−0.40	−19.58 ***
	Trust × Knowledge			−0.12	−8.72 ***
Consumer acceptance	Trust	0.47	391.98	0.28	7.48 ***
	Knowledge			0.19	5.45 ***
	Perceived benefit			0.21	5.48 ***
	Perceived risk			−0.21	−7.13 ***
	Trust × Knowledge			−0.01	−0.47

Note: *** *p* < 0.001.

**Table 7 foods-14-00669-t007:** The mediating effect of consumers’ perceived risk and perceived benefit under different levels of knowledge.

Dependent Variable	Independent Variable	Indirect Effects	Boot-SE	Boot 95% CI	
Perceived benefit	Knowledge			Upper	Lower
	Low (−1 SD)	0.12	0.02	0.07	0.16
	Medium (mean)	0.10	0.02	0.06	0.14
	High (+1 SD)	0.08	0.02	0.05	0.11
Perceived risk	Knowledge				
	Low (−1 SD)	0.08	0.01	0.05	0.09
	Medium (mean)	0.09	0.01	0.07	0.12
	High (+1 SD)	0.12	0.02	0.08	0.15

## Data Availability

The original contributions presented in the study are included in the article, further inquiries can be directed to the corresponding author.

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
