# Peer review of "The Impact of Trust on Chinese Consumers’ Acceptance of Meat Substitutes: The Mediating Role of Perceived Benefits and Perceived Risks, and the Moderating Role of Consumer Knowledge"

_foods, 2025, doi:10.3390/foods14040669_

Round 1
Reviewer 1 Report
Comments and Suggestions for Authors
An interesting, though not novel, topic investigated on a high sample with a moderate methodological approach.
Meat substitutes can always be interpreted by comparing with (processed) meat products. In the introduction, I heavily miss some insights on consumer attitudes toward traditional products both in China and in the developed world (for the latter, see, among others, https://doi.org/10.1186/s40100-023-00277-4).
There is no description of the sample regarding representativeness and it is not mentioned in the limitation part either.
Concerning the large sample size, the methodological approach is rather simple.
I also heavily miss a dedicated section for policy and managerial implications.
Author Response
Thank you very much for taking the time to review this manuscript. Please find the detailed responses below and the corresponding revisions/corrections highlighted/in track changes in the re-submitted files. We are truly grateful for the reviewers' comments. Your insights have been of great assistance to us.
Comments 1: Meat substitutes can always be interpreted by comparing with (processed) meat products. In the introduction, I heavily miss some insights on consumer attitudes toward traditional products both in China and in the developed world (for the latter, see, among others, https://doi.org/10.1186/s40100-023-00277-4).
Response 1: Thank you for pointing this out. We agree with this comment. Therefore, we have added relevant literature on the attitudes of European and Chinese consumers towards traditional meat products and meat alternatives in the introduction (Line 46) and the literature review section (Line 188) as required by the editorial staff.
Comments 2: There is no description of the sample regarding representativeness and it is not mentioned in the limitation part either.
Response 2: Agree. after receiving the feedback from the editorial department, we indeed realized that the issue of sample representativeness was not raised in the limitations section of the paper. Therefore, we specifically emphasized the limitations of sample representativeness in 5.3 (Line 408) of the discussion section.
Comments 3:Concerning the large sample size, the methodological approach is rather simple.
Response 3:Thank you for pointing this out. We agree with this comment. Therefore, after receiving the reply from the editorial staff, we attempted to use the structural equation model to address the issue that the model was a bit simplistic, as pointed out by them. However, the results of the model operation were not very satisfactory. After comprehensive consideration, we had no choice but to continue using the BOOTSTRAP regression method. Nevertheless, we are still extremely grateful for your valuable suggestions and will apply them in our future writing.
Comments 4:I also heavily miss a dedicated section for policy and managerial implications.
Response 4:Thank you for pointing this out. We agree with this comment. Therefore, in accordance with the suggestions from the editorial team, we have added countermeasures and suggestions in Sections 5.1 and 5.2 of the discussion. These suggestions are about how to enhance consumer trust (Line 408) and how to provide targeted consumer education (Line 417).

Reviewer 2 Report
Comments and Suggestions for Authors
Thank you for showing an interesting paper. Here are some points to improve the paper:
1. Theoretical Background: The current theoretical background only explains individual connections and lacks an overarching framework that integrates Figure 1. Without such integration, this section becomes an extended literature review rather than a theoretical background. It is necessary to either find a theory that supports the overall structure of Figure 1 or clearly state that this study focuses on extending connectivity through a literature review. For example, prior studies suggest that Perceived Benefit is an antecedent of Trust, and others propose Perceived Risk as an antecedent of Trust. Including these studies would likely require redrawing Figure 1 to align with the theoretical framework. A collection of literature references alone does not constitute a theoretical framework, as rearranging the references could produce different frameworks. To resolve this, identify a theory that directly or indirectly supports the current model. Otherwise, the study should be framed as purely empirical.
2. Literature Review: A dedicated literature review section is missing. It is critical to systematically present previous research perspectives on each major variable. The current "theoretical framework" section briefly mentions literature but does not provide a comprehensive review. A structured literature review is needed to strengthen the foundation of the study.
3. Sampling and WeChat Limitations: Inviting respondents via WeChat or online forums introduces sampling biases. For instance, individuals who use social media or are more familiar with WeChat are likely to dominate the sample, limiting the generalizability of the study. This approach does not align with random sampling methods and restricts the study to frequent WeChat users. The authors should describe efforts made to mitigate these limitations. If no such efforts were made, the limitations should be explicitly acknowledged.
4. Figure Typographical Error: Correct the typo in the figure: replace Preceived with Perceived.
5. Low Trust and High Trust Classification: Clarify how Low Trust and High Trust were defined. Similarly, explain the classification for Knowledge. In the analysis, both variables are continuous, but the figure implies a binary transformation. The rationale and implications of this binary conversion should be explicitly stated.
Author Response
Thank you very much for taking the time to review this manuscript. Please find the detailed responses below and the corresponding revisions/corrections highlighted/in track changes in the re-submitted files. We are truly grateful for the reviewers' comments. Your insights have been of great assistance to us.
Comments 1 : Theoretical Background: The current theoretical background only explains individual connections and lacks an overarching framework that integrates Figure 1. Without such integration, this section becomes an extended literature review rather than a theoretical background. It is necessary to either find a theory that supports the overall structure of Figure 1 or clearly state that this study focuses on extending connectivity through a literature review. For example, prior studies suggest that Perceived Benefit is an antecedent of Trust, and others propose Perceived Risk as an antecedent of Trust. Including these studies would likely require redrawing Figure 1 to align with the theoretical framework. A collection of literature references alone does not constitute a theoretical framework, as rearranging the references could produce different frameworks. To resolve this, identify a theory that directly or indirectly supports the current model. Otherwise, the study should be framed as purely empirical.
Response 1 : Thank you for pointing this out. We agree with this comment. Therefore, according to the suggestions of the editorial staff: We have further added relevant literature to consolidate the theoretical foundation, further sorted out the relationships among Trust, Knowledge, Perceived benefit, and Perceived risk, and further improved the literature review section (Line 166 187 199). We have formed a relatively complete relationship among the concepts. We have also revised the title (Line 156) and changed the theoretical framework diagram to a relationship diagram among several concepts (Line 236).
Comments 2 : Literature Review: A dedicated literature review section is missing. It is critical to systematically present previous research perspectives on each major variable. The current "theoretical framework" section briefly mentions literature but does not provide a comprehensive review. A structured literature review is needed to strengthen the foundation of the study.
Response 2: Thank you for pointing this out. We agree with this comment. Therefore, we have further strengthened the literature review section by adding literature on the relationships among various concepts and a structured literature review(Line 114).
Comments 3: Sampling and WeChat Limitations: Inviting respondents via WeChat or online forums introduces sampling biases. For instance, individuals who use social media or are more familiar with WeChat are likely to dominate the sample, limiting the generalizability of the study. This approach does not align with random sampling methods and restricts the study to frequent WeChat users. The authors should describe efforts made to mitigate these limitations. If no such efforts were made, the limitations should be explicitly acknowledged.
Response 3: Thank you for pointing this out. We agree with this comment. Therefore, after receiving the feedback from the editorial department, we indeed realized that the problem of sample representativeness was not pointed out in the limitations section of the paper. Thus, we specifically highlighted the limitations regarding sample representativeness in 5.3 (Line 455) of the discussion section.
Comments 4: Figure Typographical Error: Correct the typo in the figure: replace Preceived with Perceived.
Response 4: Therefore, based on the advice from the editorial staff, we have corrected the spelling mistakes in the figures. We sincerely apologize for this elementary error on our part. Meanwhile, we're extremely grateful for your patience and suggestions.
Comments 5: Low Trust and High Trust Classification: Clarify how Low Trust and High Trust were defined. Similarly, explain the classification for Knowledge. In the analysis, both variables are continuous, but the figure implies a binary transformation. The rationale and implications of this binary conversion should be explicitly stated.
Response 5: Thank you for pointing this out. We agree with this comment. Therefore, in accordance with the suggestions of the editorial staff, a detailed explanation of the classification of high - trust has been provided in the text (Line 358). We based on the means of the two continuous variables, trust and knowledge. The parts that are more than one standard deviation above the mean are classified as high trust and high knowledge, and the parts that are more than one standard deviation below the mean are classified as low trust and low knowledge.We chose to convert the continuous variables of trust and knowledge into binary variables mainly for the purpose of simplifying the analysis and making the relationships between variables more interpretable in certain analyses. The implications of this binary conversion are two - fold. On one hand, it provides a clear and easy - to - understand way to present the data, especially for readers who may not be familiar with more complex statistical analyses. For example, in the figure, it becomes immediately apparent how different levels of trust or knowledge are related to other variables. On the other hand, we are aware that some information may be lost in the process of converting continuous variables to binary ones. However, we believe that the benefits of simplification and enhanced interpretability in this particular stage of the analysis outweigh the potential loss of detail.

Round 2
Reviewer 1 Report
Comments and Suggestions for Authors
I went through them, and the manuscript became acceptable.
Author Response
Many thanks for your patience and suggestions. We've benefited a great deal from them.Reviewer 2 Report
Comments and Suggestions for Authors
Thank you for addressing all the previous comments well in the revised manuscript. Good luck!
Author Response
Many thanks for your patience and suggestions. We've benefited a great deal from them.